# Six-month pain and function outcome expectations were established for total knee arthroplasty using the smallest worthwhile effect

**Daniel L. Riddle**[1]*, **Nancy Henderson**[2]

**1** Departments of Physical Therapy, Orthopaedic Surgery and Rheumatology, Virginia Commonwealth University, Richmond, Virginia, United Stated of America, **2** Department of Rehabilitation Sciences, Georgia Southern University- Armstrong Campus, Savannah, Georgia, United Stated of America

* dlriddle@vcu.edu

## Abstract

### Introduction

Interpretations of patient-reported outcome measures following knee arthroplasty lack context and typically do not account for costs, risks and benefits compared to an alternative treatment. The primary purpose of our paper is to estimate expectations patients have for pain and function destination outcome, six-months following surgery relative to the outcome expected if knee arthroplasty was not done. Secondary purposes were to determine if statistically significant changes in the smallest worthwhile six-month outcome occurred following an interactive discussion and to assess the construct validity of the expected six-month outcome obtained at baseline.

### Methods

This was a secondary analysis of a prospective longitudinal cohort study of 121 patients undergoing knee arthroplasty. Smallest worthwhile effect estimates were determined and expected six-month KOOS Pain and Function, daily activity measures were established during a pre-operative visit.

### Results

The average six-month expected (the destination of interest) KOOS Pain score was 75 (IQR = 64 to 86) and the average KOOS Function, daily activity score was 74 (IQR = 59 to 86). The smallest worthwhile effect discussion led to significant changes in expected destination scores. For example, KOOS Pain expected outcome changed from 87.7 (9.8) to 75.0 (13.6), a statistically significant reduction in expected outcome ($t_{(119)}$ = 16.942, p < 0.001.

### Conclusion

Six-month expected KOOS outcomes following knee arthroplasty were established and approximate the average six-month outcomes reported in the literature. Validity of these

**Data Availability Statement:** All relevant data are within the paper and its Supporting information files. Data are available in supplemental file 3.

**Funding:** The author(s) received no specific funding for this work.

**Competing interests:** The authors have declared that no competing interests exist.

estimates was established. These data can be used to aid shared decision-making discussions regarding patient expectations of knee arthroplasty outcomes during a patient encounter.

## Introduction

The assessment of pain and function outcome following total knee arthroplasty (TKA) has been the subject of many studies [1–4]. This interest is well-placed. TKA is one of the most common and most effective major surgeries conducted worldwide [5, 6].

An as-yet unresolved issue with TKA outcome assessment is whether the journey (i.e., change from the preoperative assessment) or destination (i.e., final outcome at an established time point following surgery) should be prioritized [7]. Both can be informative but may lead to opposing conclusions. Patients with worse preoperative functional status, for example, may show greater improvements following surgery (i.e., a better journey) as compared to patients with milder symptoms but patient with milder symptoms tend to have less pain and better function following surgical recovery (i.e., a better destination) [7]. The solution to this dilemma appears to be that both journey and destination outcomes are important and that they each answer different but important questions regarding outcome following TKA.

The focus on journey outcomes has been to develop metrics to interpret the meaningfulness of changes in outcome measures following TKA [8]. The major interest has been directed to the minimal clinically important difference (MCID) family of measures [9, 10]. We have argued against the scientific soundness and utility of MCID measures [11, 12] and used the smallest worthwhile effect (SWE) of TKA as a more scientifically grounded alternative [13]. The SWE method requires patients, prior to their TKA, to estimate the outcome they believe makes TKA worthwhile. In our study, the outcome time point for the SWE was six-months following surgery. Patients were provided with estimates of the costs, risks and benefits of TKA as well as the likely outcome if TKA was not conducted. Comparison to an alternative treatment (i.e., no surgery) allows for a counterfactual—the effect of an intervention (TKA) compared to an alternative (no surgery, in our study). Because MCID-based measures are determined from observational studies, changes captured may have occurred for reasons other than true treatment effects, such as natural history, and regression to the mean [14]. In other words, MCIDs reflect changes not necessarily attributable to the treatment (i.e., the TKA). By comparing to an alternative treatment, the SWE approximates the true treatment effect that patients expect, in a way that is similar to randomized clinical trials, to estimate true treatment effects of the intervention of interest [14, 15]. By determining patient expectations in this more formal way, it is our view that surgeons, other clinicians, and patients may benefit from a more thorough determination of expectations in the context of key variables including costs, risks and benefits, relative to no TKA. Determining expectations in this way may allow for a more accurate estimation of patient outcome expectations and enhance satisfaction, an important outcome for both surgeons and patients.

Our prior paper focused on the benefits (i.e., journey-based improvements) needed to justify TKA over not having the surgery [13]. The current paper examines the six-month outcome (i.e., destination-based outcome) patients expect over that expected with no surgery, assuming that TKA is worthwhile to them. To our knowledge, this is the first study to examine patients' expectations of TKA destination outcomes compared to the outcome they would likely have if surgery was not conducted (i.e., the counterfactual). The primary purpose of the current study

is to report on the expectations patients have for pain and function destination outcome, six-months following surgery relative to the outcome expected if TKA was not done. A secondary purpose was to determine if statistically significant changes in the smallest worthwhile destination outcome occurred following the SWE discussion. Finally, to assess the construct validity of the expected destination outcome obtained at baseline, we compared participants who either did or did not achieve their expected destination outcome at the six-month follow-up to a satisfaction rating (yes or no) also obtained six months following surgery.

## Materials and methods

### Participant recruitment

This was a secondary analysis of a prospective longitudinal study patients scheduled for a unilateral TKA for osteoarthritis at a hospital in Savannah, Georgia, USA. All adult patients aged 45–90 years, diagnosed with symptomatic knee osteoarthritis (OA), and scheduled for primary TKA by one of nine participating surgeons were considered for inclusion, regardless of race or gender. Participants were excluded if they were scheduled for revision TKA, simultaneous bilateral knee arthroplasty, unicompartmental knee arthroplasty, or knee arthroplasty for cancer or rheumatic disease. A total of 10 patients were excluded because they met one or more exclusion criteria. A total of 250 met inclusion criteria: 129 declined participation after and 121 provided written informed consent and were included in this study.

### Procedures

Participants meeting our criteria were recruited prior to or following their required preoperative TKA educational class. A prior paper provides a complete description of the methods for data collection [13]. Following freely-given written informed consent approved by the Institutional Review Boards at Virginia Commonwealth University and the participating hospital, participants completed baseline questionnaires. Study recruitment occurred from January 15, 2018 to January 10, 2019. One of us, (NH) interviewed each participant to determine the SWE for KOOS Pain and KOOS Function, daily living scales. The SWE required participants to express their expectations regarding their outcomes following TKA and this was the novel and important component in our study. KOOS Pain is a validated 9-item pain with activity scale while KOOS Function, daily living is a validated 17-item difficulty-with-activity scale [16, 17]. Both scales ranged from 0 to 100 with higher scores equating to less pain or better function. Participants were instructed to complete the KOOS scales based on their current knee pain and functional status. We chose the KOOS Pain and Function, daily living scales because these are two of the most commonly recommended and validated scale for patients with knee osteoarthritis and arthroplasty. Both were derived directly from the WOMAC scale and, at the time of this study, were available for public use [16, 17].

Next, participants were asked to complete a second set of KOOS Pain and Function, daily living scales that represented their "initial expected" six-month outcome, considering the 10% to 15% average worsening expected without TKA, and given the costs and risks of surgery [18, 19]. The standardized script (see S1 File) included estimates of costs, and risks of TKA [20, 21] as well as likely outcomes for the alternative treatment of no TKA [19].

After the "initial expected" six-month KOOS scales were completed, NH determined whether the expected six-month outcome scales actually represented the SWE as compared to not having TKA. The investigator did this by first identifying the item with the lowest (least) pain (or function) rating and systematically increasing the score on the item by one. For example, if the participant marked "no pain" for KOOS Pain item #3, "Straightening knee fully" the participant was asked if the effect of TKA would still be worthwhile, relative to no TKA, if the

score was changed by one point to "mild pain." This process was repeated with the next lowest scored item until the participant indicated that the surgery would no longer be worthwhile as compared to not having the surgery. The expected six-month KOOS Pain and Function, daily living scores identified after this iterative process were termed the "final expected" six-month score. The current investigation focused on the "initial expected" and "final expected" six-month KOOS Pain and Function, daily living scores as well as the actual six-month KOOS Pain and Function, daily living scores, collected six-months after surgery.

Prior work using the same dataset focused on change from the preoperative visit to the 6-month postoperative visit (i.e. the journey). The current paper focuses specifically on 6-month (i.e. destination) pain and function patient-reported outcome. Estimates of both the magnitude of change from preoperative visit to final outcome at a set time point and final outcome are important to patients and are driven by different factors, including satisfaction and expectations [22–25]. Our focus on 6-month outcomes is driven by evidence indicating that 6-month outcomes capture approximately 90% or more of the benefit associated with TKA [4]. Additionally, some patients and surgeons may better relate to an understanding of typical destination outcome as compared to journey change scores leading patients and clinicians to benefit from access to both journey and destination SWE estimates.

### Baseline variables

Age in years and biological sex were recorded. Depressive symptoms were quantified using the PHQ-8 depressive symptom screening scale [26, 27], a validated measure ranging from 0 (no depressive symptoms) to 24 (eight depressive symptoms nearly every day). Pain catastrophizing was measured with the Pain Catastrophizing scale, a validated 16-item instrument ranging from 0 (no pain catastrophizing) to 52 (severe catastrophizing) [28]. Anxiety was measured with the GAD-7 [29], a validated anxiety screening tool ranging from 0 (no anxiety to severe anxiety nearly every day. Arthritis self-efficacy was measured with the validated Arthritis Self-Efficacy Scale, a validated measure of a participant's degree of certainty in being able to complete a variety of tasks [30]. The scale ranges from 8 (very uncertain about being able to do the tasks) to 80 (very certain about being able to do the tasks).

### Six-month postoperative KOOS measurements

Participants were contacted via telephone (with up to 6 attempts) by NH to determine their actual six-month postsurgical KOOS Pain and Function, daily activity scores. Participant satisfaction with outcome was determined by use of the Patient Acceptable Symptom State (PASS) questionnaire [31]. The single item PASS question was ""Taking into account all the activities you have during your daily life, your level of pain, and also your functional impairment, do you consider that your current state is satisfactory?" [31].

### Data analysis

Our study was originally powered to detect a 50th percentile estimate for the SWE. A sample size of 120 would result in a 95% confidence interval of 0.40 to 0.60 which we believed to be reasonable for estimating the 50th percentile for the SWE for the average patient. The study was not powered to establish precision for the six-month postoperative KOOS measures reported on in the current study because it was the baseline scores that were most critical for SWE estimation. While estimates of sample size for the current study were not informed by prior work, the sample size is similar to prior SWE studies on patients with a variety of diagnoses [32–34].

We report 50th and 90th percentiles, marked on histograms, displayed "average patient" final expected destination scores (50th%ile). Scores needed by the great majority of patients (90th%ile) to consider their six-month outcome as worthwhile, as compared to not having surgery were also determined [13, 32].

To determine if "final expected" six-month scores were significantly different from "initial expected" six-month scores, we used paired sample t-tests to compare scores for KOOS Pain and KOOS Function, daily activity scores. Pearson chi square tests were used to compare actual six-month postsurgical PASS scores (yes or no) and achievement of "final expected" six-month KOOS Pain and KOOS Function, daily activity scores at the six-month follow-up. We used SPSS version 28.0.1.1 and a p value of ≤ 0.05 for all analyses.

## Results

Our sample consisted of 121 participants, 54% were women and the average age of the sample was 67 (sd = 9.7) years. Baseline mean scores were 44.2 (sd = 16.9) for KOOS Pain scores and 46.0 (sd = 19.1) for KOOS Function, daily activity (see Table 1). A total of 82 participants completed six-month follow-up KOOS questionnaires (68% of the sample). To assess risk of bias attributable to missing data, we examined for differences in baseline scores between those who did (n = 82) and did not (n = 39) complete the 6-month follow-up. Independent samples t-tests for continuous variables and Pearson Chi Square tests for categorical variables were calculated to assess potential biased loss to follow-up. We found no statistically significant differences (See S2 File). In a second risk of bias analysis, we excluded participants with missing 6-month data and found essentially no differences in "final expected" KOOS 50th %ile and

**Table 1. Baseline participant characteristics.**

| Sample Characteristics | N = 121 | Missing |
|---|---|---|
| Age (yr), mean (sd) | 67.7 (7.6) | 3 |
| Sex (women), n (%) | 64 (54) | 2 |
| Baseline KOOS* Pain Score, mean (sd) | 44.2 (16.9) | 0 |
| Initial Expected KOOS Pain at 6-months, mean (sd) | 87.7 (9.8) | 1 |
| Final Expected KOOS pain at 6-months, mean (sd) | 74.9 (13.6) | 1 |
| Baseline KOOS Function, daily living^, mean (sd) | 46.0 (19.1) | 1 |
| Initial Expected 6-month KOOS Function, mean (sd) | 82.3 (12.9) | 1 |
| Final Expected KOOS Function at 6-months, mean (sd) | 72.1 (17.5) | 1 |
| Actual 6-month KOOS Pain, mean (sd) | 90.2 (14.4) | 39 |
| Actual 6-month KOOS Function, mean (sd) | 92.4 (12.8) | 40 |
| PHQ-8#, mean (sd) | 4.9 (5.3) | 2 |
| GAD-7+, mean (sd) | 4.8 (4.8) | 2 |
| Pain Catastrophizing Scale, mean (sd) | 16.1 (13.3) | 1 |
| Self-efficacy, mean (sd) | 31.4 (18.6) | 1 |

*Knee Injury and Osteoarthritis Score (KOOS) pain subscale- baseline pain (0–100 with lower numbers indicating greater pain)

^Knee Injury and Osteoarthritis Score (KOOS) function subscale- baseline function (0–100 with lower numbers indicating greater difficulty with functional activities)

#Personal Health Questionnaire (PHQ-8) Depression Scale- depressive feelings (0–24 with higher numbers indicating greater feelings of depression)

+Generalized Anxiety Disorder (GAD)- baseline anxiety (0–21 with higher numbers indicating greater feelings of anxiety)

**Table 2. The 50th and 90th percentile destination smallest worthwhile effect scores for KOOS Pain and KOOS Function, daily activity scores.**

| Measure | 50th percentile score (IQR) | 90th percentile score |
|---|---|---|
| **KOOS Pain*** | 75 (63 to 86) | 90 |
| **KOOS Function, daily activity^** | 74 (59 to 86) | 94 |

*Knee Injury and Osteoarthritis Score (KOOS) pain subscale- baseline pain (0–100 with lower numbers indicating greater pain)

^Knee Injury and Osteoarthritis Score (KOOS) function subscale- baseline function (0–100 with lower numbers indicating greater difficulty with functional activities)

IQR = Interquartile range

90th %ile outcomes as compared to the entire sample (data not provided). Raw data are provided in S3 File.

## SWE estimates

Histograms with 50th %iles and 90th %iles for KOOS Pain and KOOS Function, daily activity scores for the complete sample appear in Table 2 and Fig 1. The average patient, reflected by the 50th %ile score was 75 points for KOOS Pain (IQR = 63 to 86) and 74 points for KOOS Function, daily activity (IQR = 59 to 86). The 90th %ile, reflecting expected outcomes for the great majority of patients, was 90 points for KOOS Pain and 94 points for KOOS Function, daily activity (see Table 2 and Fig 1).

## Expectations assessment

The paired sample t-test comparing "initial expected" to "final expected" six-month KOOS Pain scores indicated that these scores were significantly different. For KOOS Pain, the mean (sd) score for "initial expected" six-month outcome was 87.7 (9.8) and for final expected KOOS Pain, the mean was 75.0 (13.6). This difference was highly significant (t (119) = 16.942, $p < 0.001$). For KOOS Function, daily activity, the mean (sd) score for "initial expected" six-month outcome was 82.3 (12.9) and for "final expected" KOOS Function, daily activity, the mean was 72.1 (17.5). This difference also was highly significant (t (119) = 10.842, $p < 0.001$).

## Construct validity of SWE estimates

The Pearson chi square test comparing PASS satisfaction scores (yes or no) obtained at the six-month postsurgical time point and actual achievement of "final expected" six-month KOOS Pain was highly significant $X^2$ (1, N = 82) = 25.469, $p < 0.001$. A total of 42.9% (6/14) of participants whose actual six-month KOOS Pain score was less than their final expected KOOS Pain score were dissatisfied with their outcome while 1.5% (1/68) of participants who scored equal to or better than their final expected KOOS Pain score were dissatisfied with their outcome. Similarly, the Pearson chi square test for KOOS Function was also significant with $X^2$ (1, N = 82) = 23.479, $p < 0.001$. A total of 50% (4/8) of participants whose actual six-month KOOS scores were less than their final expected KOOS Function, daily activity scores were dissatisfied with their outcome. A total of 2.7% (2/73) of participants who scored equal to or better than their final expected KOOS Pain score were dissatisfied with their outcome.

A

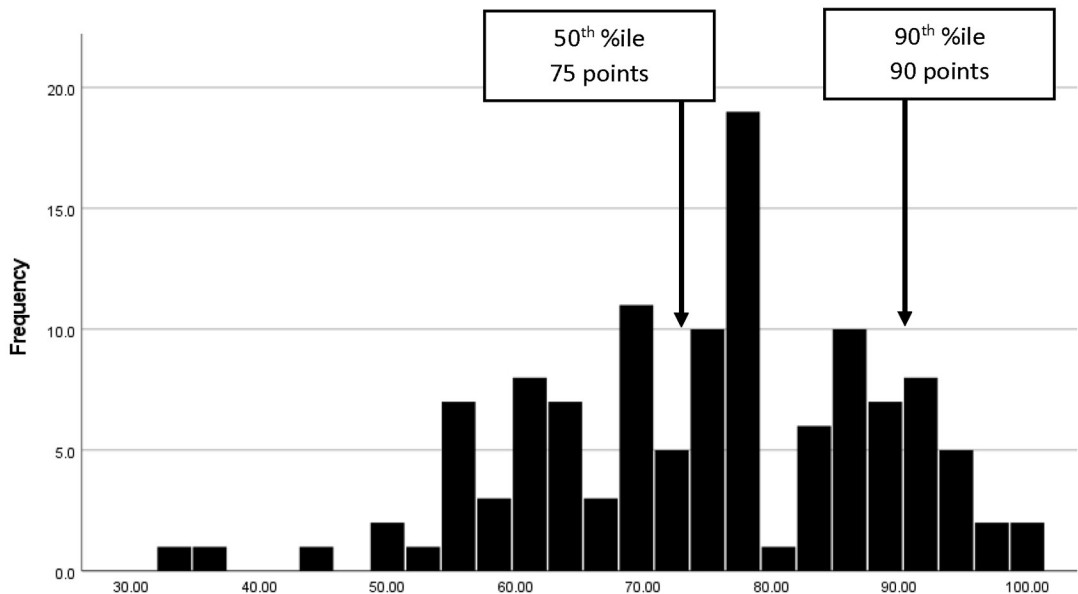

B

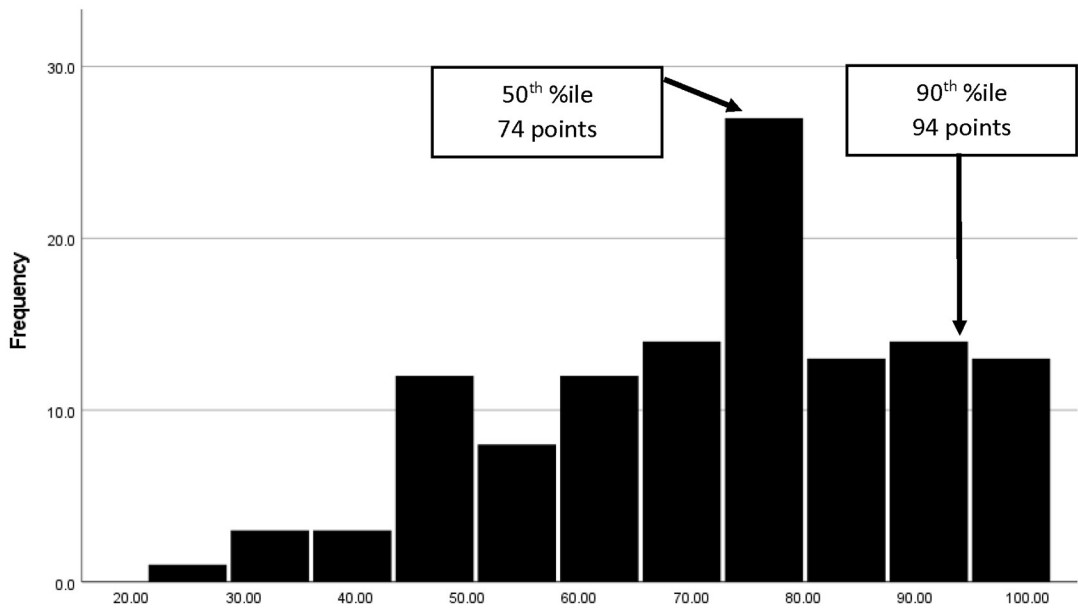

**Fig 1. Distributions of six-month destination outcomes.** The panels illustrate the distributions of six-month destination outcomes based on the Smallest Worthwhile Effect (SWE) procedures for both KOOS Pain (panel A) and for KOOS Function, daily activity (panel B). Both panels illustrate the location and score for the 50th percentile (the average patient) and 90th percentile (the great majority of patients).

## Discussion

Several important findings were reported in our study. First, using a scientifically defensible method for assessing patient destination outcome expectations, the average patient expects scores of 75 for KOOS Pain and 74 for KOOS Function, daily activity, six months following TKA. These scores are very similar to actual six-month average KOOS outcomes reported in a recently published systematic review of the clinical course of pain and function following TKA [4]. Second, the method applied to determine destination outcome led to statistically significant reductions in initial six-month outcome expectations. Patients appear to initially estimate a larger than required improvement when asked to estimate the needed effect to justify TKA given costs and risks as compared to not having the surgery. Using follow-up questions about specific KOOS items, patients adjusted their KOOS scores to indicate lower (easier to achieve) expectations relative to the initial scores. This approach allows for a more realistic expectation estimate and provides a method for surgeons and their staff to capture a more accurate estimate of needed outcome during a patient encounter prior to surgery. Third, this more realistic estimate is strongly associated with actual six-month outcome satisfaction scores. Patients who meet or exceed preoperative expectations with TKA are very likely to be satisfied with their outcome as compared to patients whose outcome falls short of initial expectations [35, 36]. Patients who fall short of baseline expectations at the 6-month postoperative time point were significantly more likely to be dissatisfied with their outcome (i.e., approximately 50% dissatisfaction rate versus an estimated 2% dissatisfaction rate for those meeting or exceeding expectations). Our study supports the method we used to determine expectations as a valid estimate of patients' true preoperative expectations of destination outcome. Questioning about expectations in the way we did is novel and a highly important component to our study. This new evidence, in total, supports the validity of SWE-based methods for determining destination outcome following TKA.

Expectations of outcome following TKA, while challenging to conceptualize and measure [37] are associated with outcome following surgery [35, 36, 38, 39]. Hawker and colleagues, for example, have included outcome expectation assessment in judging whether TKA should be offered to patients [40, 41]. Randomized trials also have been designed to modify surgical outcome expectations in TKA [42, 43]. Evidence supports inclusion of outcome expectation assessment during the shared decision-making discussion with patients and our data suggest that six-month expectations similar to those reported in our study appear to be realistic relative to actual outcome. Additionally, if patients meet or exceed these estimates, they are extremely likely to be satisfied with their destination outcome.

One approach for inclusion of destination outcome discussions with patients is by use of a KOOS score map. Fig 2 illustrates the recently published KOOS Pain score map [44] with inclusion of a line indicating the six-month expected destination outcome score for the average patient. During the shared decision making (SDM) discussion, the surgeon could review the score map with a patient to show average pre-operative and one-year post-operative scores (see Fig 2) as well as the average destination score patients expect six-months following surgery. If the patient's expectation for six-month outcome exceeded this score, the surgeon could discuss whether the expectation was realistic, given the patient's current status.

To further inform interpretation of our findings, we examined final estimated six-month destination scores by baseline KOOS Pain tertile to determine if baseline KOOS Pain score influenced the expected destination score. Baseline patient-reported outcome scores generally have been found to predict not only the magnitude of change in outcome score [45, 46] but also the actual destination outcome score [47]. We found that expected destination outcome for participants in the lowest tertile of baseline KOOS score (worse pain and functional

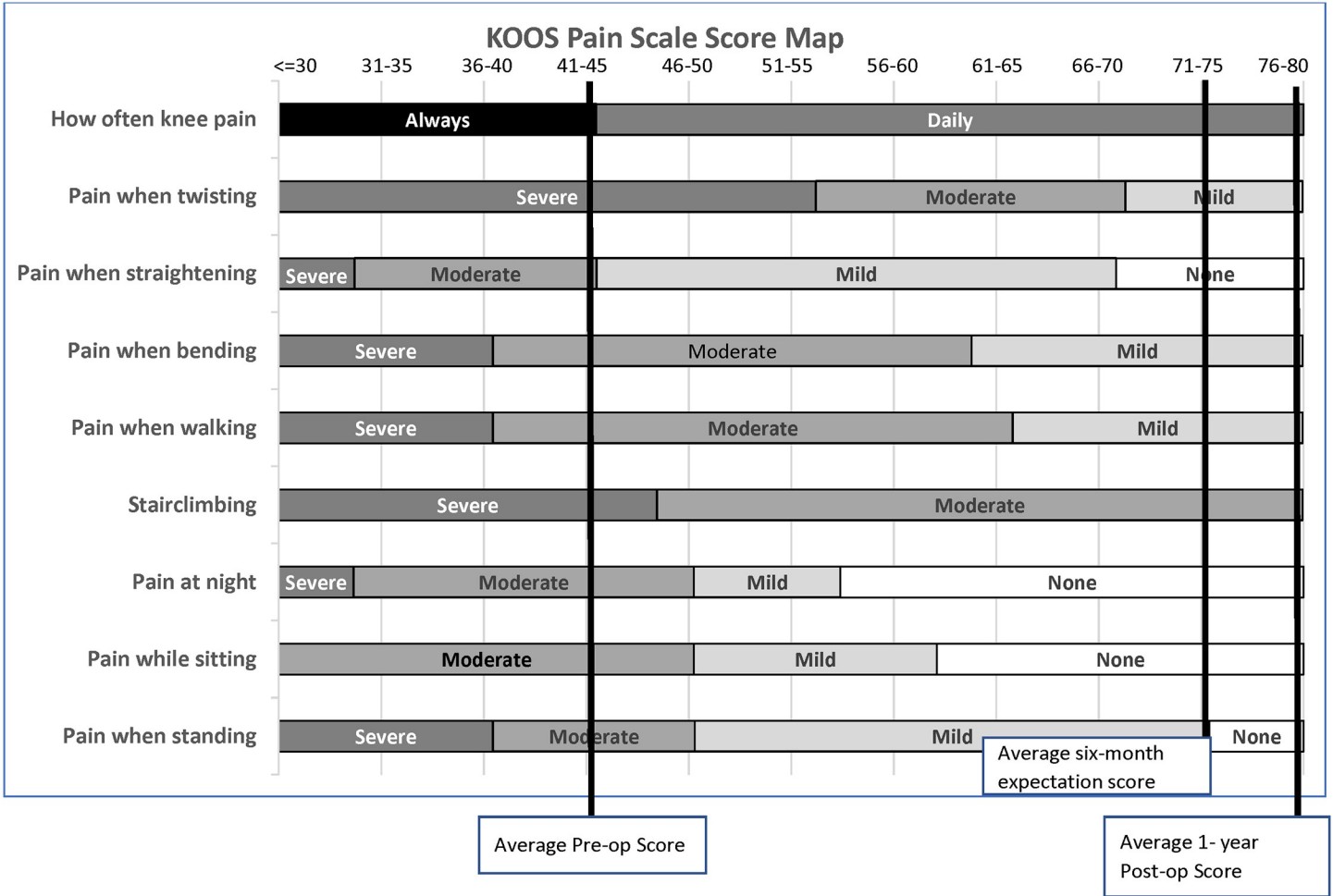

**Fig 2. KOOS Pain score map.** KOOS Pain scale score map with marked vertical lines representing evidence-derived average preoperative, average 1-year postoperative and estimated 6-month postoperative expected scores.

difficulty) had lower destination expectations as compared to participants in the middle or highest (i.e., least painful or least functional difficulty) tertiles (see Table 3). This finding also is consistent with prior evidence and suggests that baseline KOOS score associates with destination expectation score, particularly for patients with more severe pain or functional loss.

Our study has several strengths including a prospective design, reasonably large sample size for the type of study conducted, and the novelty of the methods. There also were some

**Table 3. 50th percentile destination outcome estimates for KOOS Pain and KOOS Function, daily living scales stratified by baseline tertile score.**

| Scale and Destination Estimate | Tertile (score range) | | |
|---|---|---|---|
| | Score (sd) | Score (sd) | Score (sd) |
| **KOOS Pain** | 1st tertile (0–36) | 2nd tertile (36.1–50) | 3rd tertile (>50) |
| **Destination estimates, 50th%ile** | 70.0 (15.6) | 75.3 (12.3) | 78.2 (12.7) |
| **KOOS Function, daily living** | 1st tertile (0–37) | 2nd tertile (37.1–50.5) | 3rd tertile (>50.5) |
| **Destination estimates, 50th%ile** | 62.2 (18.1) | 76.4 (13.2) | 76.9 (17.1) |

weaknesses. Loss to follow-up is a challenge in many longitudinal studies and also was a concern in the current study. While 32% of the sample had missing six-month postsurgical data, the missing data analysis indicated no bias, though it is possible that unmeasured confounding still influenced the findings. There was substantial variation in patient estimates of six-month destination outcomes. This variation is not surprising given that patients have widely varying reasons for seeking out TKA, different preoperative and early postoperative expectations and different comorbidities, baseline pain and functional status complaints [24, 25]. Variation in pre-operative determinations of destination outcomes, however, does create challenges in applying destination outcome estimates from this study to clinical practice. Not every patient is reflected in the "average patient" score. Use of the tertiles reported in Table 2 may help to offset this limitation. We had 129 participants decline participation after determining that they met all inclusion criteria and this may have impacted generalizability of the findings if characteristics of these participants differed from the consented sample. Additionally, our sample was generally well educated with relatively high income and this may have biased the results. We did not specifically determine participants' chief complaints but it is likely that the great majority sought TKA because of knee pain and compromised function, and both are captured by KOOS scale scores [48]. Finally, the current study was not powered to detect a smallest worthwhile benefit because no studies were found to inform a power analysis. With this said, we found statistically significant changes in final outcome expectations compared to initial outcome expectations after the SWE method was applied.

## Conclusion

The SWE method for determining six-month postoperative destination outcome during a preoperative visit demonstrated acceptable validity and may assist surgeons and patients during shared decision-making discussion regarding likely outcomes following TKA. Average six-month destination outcomes for KOOS Pain and KOOS Function, daily living were similar to actual six-month outcome and can be used to assist in the formulation of realistic expectations during the patient/surgeon encounter. The score map figure presented in this study may provide a useful graphic for facilitating TKA outcome expectation discussions.

## Supporting information

**S1 File. Script used for data collection.**
(DOCX)

**S2 File. Missing data analysis.**
(DOCX)

**S3 File. Raw data file.**
(XLSX)

**S1 Checklist. Human participants research checklist.**
(DOCX)

## Author Contributions

**Conceptualization:** Daniel L. Riddle.

**Data curation:** Nancy Henderson.

**Formal analysis:** Daniel L. Riddle.

**Investigation:** Daniel L. Riddle.

**Methodology:** Daniel L. Riddle.

**Project administration:** Nancy Henderson.

**Writing – original draft:** Daniel L. Riddle.

**Writing – review & editing:** Daniel L. Riddle, Nancy Henderson.

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
