## [Decision Letter · Decision Letter 0]

8 Dec 2023

PONE-D-23-35725Six-Month Pain and Function Outcome Expectations Were Established for Total Knee Arthroplasty Using the Smallest Worthwhile EffectPLOS ONE

Dear Dr. Riddle,

Thank you for submitting your manuscript to PLOS ONE. After careful consideration, we feel that it has merit but does not fully meet PLOS ONE’s publication criteria as it currently stands. Therefore, we invite you to submit a revised version of the manuscript that addresses the points raised during the review process.

We look forward to receiving your revised manuscript.

Kind regards,

Hassan Zmerly, MD PhD

Academic Editor

PLOS ONE

Reviewers' comments:

Reviewer's Responses to Questions

**Comments to the Author**

1. Is the manuscript technically sound, and do the data support the conclusions?

Reviewer #1: Yes

Reviewer #2: Yes

2. Has the statistical analysis been performed appropriately and rigorously? 

Reviewer #1: Yes

Reviewer #2: Yes

3. Have the authors made all data underlying the findings in their manuscript fully available?

Reviewer #1: No

Reviewer #2: No

4. Is the manuscript presented in an intelligible fashion and written in standard English?

Reviewer #1: Yes

Reviewer #2: Yes

5. Review Comments to the Author

Reviewer #1: I will specify :

1) results are not well reported in a table(supplemental file deals only with baseline findings)

2)line97: misses the number of participants before exclusions and denials

3)line98: there is no reference to baseline daily activity level and pain at start of observation or any link to the table.

4)line 114: why did you prefer to use KOOS instead of other scale?This is not explained in the introduction or in the methods

5)line168: the sample size should be 121 while it is written 120

6)in the discussion & conclusions(from line 242): you did not refer to selection bias even though your sample is mainly well educated and with good income.This could influence the understanding of the questionnaire and the aim of the study with a rejection response higher from people with a lower education who usually have worst outcome after 6 months.

7)line 235: the word "was" is repeated twice(TYPO)

Reviewer #2: Overall Comment: This is a well-written manuscript that provides important information about patient expectations and outcomes. It is relevant to the PT and ortho community and is suitable for publication.

Please find below some minor modifications to further improve this work.

Comment #1: The manuscript describes a technically sound method for determining 6-month post outcomes' expectations during pre-op in patients undergoing TKA. The method and the data support this conclusion.

However, I would recommend making the distinction/clarification and mentioning the "expectation" aspect of this throughout the manuscript i.e. lines 110-111, 143, 260-261, 318. The expectation is the most important nuance in this manuscript.

Comment #2: Introduction

- In the introduction, lines 68-79: this explains the SWE method and what it requires but also what and how it was used in this study. This should be moved to material and methods rather than the intro. Please consider adjusting accordingly.

- Add a few sentences in the intro about the importance that finding the SWE could have clinically for surgeons and for patients, with references, if possible.

Comment #3: Materials and Methods

- Lines 94-95: consider adding a subheading called Recruitment or Study Population or something like that since you are using subheadings for the rest of this section.

- line 95: you mention 121 persons, better use "patients"

- line 97: "men and women", some people are non-binary etc. I would say "all adult patients aged 45-90, diagnosed with symptomatic knee osteoarthritis (OA), and scheduled for primary TKA by one of our nine participating surgeons were considered for inclusion in this study regardless of race or gender."

- line 95, you mention 121 persons. then say 10 were excluded (line 101), then say 129 met criteria but declined (line 102) and you ended up with 121 patients. While I understand this, it would be nicer and easier to the reader to see the breakdown of those number. Something like the following:

"This was a secondary analysis of a prospective longitudinal study patients scheduled for a unilateral TKA for osteoarthritis at a hospital in Savannah, Georgia, USA. All adult patients aged 45-90, diagnosed with symptomatic knee osteoarthritis (OA), and scheduled for primary TKA by one of our nine participating surgeons were considered for inclusion in this study regardless of race or gender. Participants were excluded if they were scheduled for revision TKA, simultaneous bilateral knee arthroplasty, unicompartmental knee arthroplasty, or knee arthroplasty 100 for cancer or rheumatic disease. A total of 10 patients were excluded because they met one or more exclusion criteria. A total of 250 met inclusion criteria: 129 declined participation after and 121 provided written informed consent and were included in this study."

- Apart from baseline characteristics, it is important to add the chief complaints of patients/the reason for TKA. As this alone might alter the expectation, pain and outcomes...

- line 108: would omit Virgina Commonwealth University. Replace by "the university Institutional Review Boards".

- line 118: please if possible cite a source for the "10% to 15% average worsening expected without TKA" as this is a big claim.

- line 124-127: excellent explanation

- line 146-147: annual income and educational level are mentioned. Consider explaining why these were collected/if looking at SES, etc.

- line 162: regarding the PASS, it remains unclear to the reader, without going back to the reference, whether the PASS only consists of one question (lines 163-165) or it it's a full questionnaire (like mentioned in line 163). Please clarify.

Comment #4: Data analysis

- line 167: why was the study powered to detect a 50th percentile estimate only. Why not 75th as per the KOOS measure in other publications. Please justify/clarify.

- line 169-171: why was the study not powered?

- lines 177-182: the correct tests were chosen and the statistical analyses serve the purpose of this study.

Comment #5: Results

- For clarity purposes, consider using subheadings for the results either as per the materials and methods or as per the outcomes.

- Line 188: specify that it's the KOOS questionnaires.

- Line 195: good justification, please provide the data as supplemental file #3.

- line 236-237: weird sentence, please review.

Comment #6: Discussion

- line 245: define KA as it has not been used before in the manuscript, or use TKA which has. Adjust manuscript accordingly.

- line 250-254: excellent and relevant

- line 256/269: it would be interesting to talk about the underestimation of estimates too. I might have misunderstood but it's also unclear how overestimation lead to better satisfaction (line 269) rather than lesser. It would be nice to discuss all three outcomes, underestimation, adequate estimation and overestimation.

- line 274: define SDM

Comment #7: Conclusion

- Adequate, answers the research question and provides solutions.

Well done overall! Very interesting.

6. PLOS authors have the option to publish the peer review history of their article (what does this mean?). If published, this will include your full peer review and any attached files.

Reviewer #1: **Yes: **Valentina Di Gregori

Reviewer #2: No

---

## [Author Response · Author response to Decision Letter 0]

15 Dec 2023

Reviewers' comments:

Reviewer's Responses to Questions

We are grateful to the review team for the thorough review of our paper. In response to the comments, we have highlighted the changes in red.

Comments to the Author

1. Is the manuscript technically sound, and do the data support the conclusions?

Reviewer #1: Yes

Reviewer #2: Yes

2. Has the statistical analysis been performed appropriately and rigorously?

Reviewer #1: Yes

Reviewer #2: Yes

3. Have the authors made all data underlying the findings in their manuscript fully available?

Reviewer #1: No

Reviewer #2: No

4. Is the manuscript presented in an intelligible fashion and written in standard English?

Reviewer #1: Yes

Reviewer #2: Yes

Author responses to general questions: We very much appreciate the helpful feedback and guidance from the review team. We have addressed each comment below and have made what we believe to be substantive responses to each query by highlighting the changes in red. 

5. Review Comments to the Author

Reviewer #1: I will specify :

1) results are not well reported in a table(supplemental file deals only with baseline findings)

Author response: Thank you for the suggestion. We have added a Table 2 to succinctly summarize the key findings in a Table. 

Changes to manuscript: We have added a Table 2 with key findings reported in Table format. Please see page 10, lines 230-239.

2)line97: misses the number of participants before exclusions and denials

Author response: Thank you. The total sample screened and excluded was already fully described a few lines below this. 

Changes to manuscript: No changes made. 

3)line98: there is no reference to baseline daily activity level and pain at start of observation or any link to the table.

Author response: The baseline KOOS pain and Function, daily activity scores are reported in the first few lines of the Results section as well as the original Table 1 and we believe that this is the appropriate place to report this information.

Changes to manuscript: No changes made. 

4)line 114: why did you prefer to use KOOS instead of other scale?This is not explained in the introduction or in the methods

Author response: Thank you for the comment. We now briefly provide a justification for the use of KOOS, one of the most commonly recommended and validated measures in osteoarthritis and arthroplasty. 

Changes to manuscript: Please see lines 124-127 on page 6. “We chose the KOOS Pain and Function, daily living scales because these are two of the most commonly recommended and validated scale for patients with knee osteoarthritis and arthroplasty. Both were derived directly from the WOMAC scale and, at the time of this study, were available for public use [16,17]. “ 

5)line168: the sample size should be 121 while it is written 120

Author response: Thank you. The number of 120 was used in a power analysis and does not reflect the total sample recruited. The sample size calculation used an n = 120 so this is the correct number. 

Changes to manuscript: No changes required. 

6)in the discussion & conclusions(from line 242): you did not refer to selection bias even though your sample is mainly well educated and with good income.This could influence the understanding of the questionnaire and the aim of the study with a rejection response higher from people with a lower education who usually have worst outcome after 6 months.

Author response: Excellent point. We have added this to our limitations section as a potential source of bias. 

Changes to manuscript: Please see lines 342-343 on page 15. “Additionally, our sample was generally well educated with relatively high income and this may have biased the results.”

7)line 235: the word "was" is repeated twice(TYPO)

Author response: Thank you. The typo was corrected. 

Changes to manuscript: Please see line 261, page 11.

Reviewer #2: Overall Comment: This is a well-written manuscript that provides important information about patient expectations and outcomes. It is relevant to the PT and ortho community and is suitable for publication.

Please find below some minor modifications to further improve this work.

Author response: We very much appreciate your positive comment endorsing the importance of our paper. 

Comment #1: The manuscript describes a technically sound method for determining 6-month post outcomes' expectations during pre-op in patients undergoing TKA. The method and the data support this conclusion.

However, I would recommend making the distinction/clarification and mentioning the "expectation" aspect of this throughout the manuscript i.e. lines 110-111, 143, 260-261, 318. The expectation is the most important nuance in this manuscript.

Author response: Thank you for this important point. We have emphasized the novelty and importance of this issue in multiple areas. 

Changes to manuscript: Please see lines 118-120 on page 5 “The SWE required participants to express their expectations regarding their outcomes following TKA and this was the novel and important component in our study. “ and lines 289-90 on page 13 “ Questioning about expectations in the way we did is novel and a highly important component to our study. “

Comment #2: Introduction

- In the introduction, lines 68-79: this explains the SWE method and what it requires but also what and how it was used in this study. This should be moved to material and methods rather than the intro. Please consider adjusting accordingly.

Author response: Given the importance of the SWE method and it’s key role in our study, we actually think it is important to emphasize it in the introduction to make the reader aware of the role of SWE and expectation assessment in our study. Given the novelty of this approach we think it’s quite important to make the reader aware of this method in a brief and general way before providing more detailed description in the Methods. This section also is written in a way that provides a more general description of this new approach, an issue that we believe to be important enough that the reader should have some sense of this method before getting to the detail. We have added some elaboration to guide the reader on why we believe this aspect of the study to be important. 

Changes to manuscript: Please see lines 79-84 on page 4. “By determining patient expectations in this more formal way, it is our view that surgeons, other clinicians, and patients may benefit from a more thorough determination of expectations in the context of key variables including costs, risks and benefits, relative to no TKA. Determining expectations in this way may allow for a more accurate estimation of patient outcome expectations and enhance satisfaction, an important outcome for both surgeons and patients. “

- Add a few sentences in the intro about the importance that finding the SWE could have clinically for surgeons and for patients, with references, if possible.

Author response: We appreciate the suggestion and believe this adds important information to the general description already included in the introduction and addressed in the comment above. 

Changes to manuscript: This point overlaps with the point above. Please see lines 79-84 on page 4. “By determining patient expectations in this more formal way, it is our view that surgeons, other clinicians, and patients may benefit from a more thorough determination of expectations in the context of key variables including costs, risks and benefits, relative to no TKA. Determining expectations in this way may allow for a more accurate estimation of patient outcome expectations and enhance satisfaction, an important outcome for both surgeons and patients. “

Comment #3: Materials and Methods

- Lines 94-95: consider adding a subheading called Recruitment or Study Population or something like that since you are using subheadings for the rest of this section.

Author response: Thank you. A new heading has been added. 

Changes to manuscript: Subheading added to Materials and Methods section on page 4. 

- line 95: you mention 121 persons, better use "patients"

Author response: Thank you for the suggestion. The standard approach when describing patients recruited for a research study is to consistently describe them as “participants.” We now consistently use the word participants. The description in our paper makes clear that our participants were patients at the time of the study. 

Changes to manuscript: Changed to “participants” throughout the paper when referring specifically to the participants in our study. 

- line 97: "men and women", some people are non-binary etc. I would say "all adult patients aged 45-90, diagnosed with symptomatic knee osteoarthritis (OA), and scheduled for primary TKA by one of our nine participating surgeons were considered for inclusion in this study regardless of race or gender."

Author response: Thank you. We have made this change. 

Changes to manuscript: Please see line103-104 on page 5. “All adult patients aged 45-90 years,…”

- line 95, you mention 121 persons. then say 10 were excluded (line 101), then say 129 met criteria but declined (line 102) and you ended up with 121 patients. While I understand this, it would be nicer and easier to the reader to see the breakdown of those number. Something like the following:

"This was a secondary analysis of a prospective longitudinal study patients scheduled for a unilateral TKA for osteoarthritis at a hospital in Savannah, Georgia, USA. All adult patients aged 45-90, diagnosed with symptomatic knee osteoarthritis (OA), and scheduled for primary TKA by one of our nine participating surgeons were considered for inclusion in this study regardless of race or gender. Participants were excluded if they were scheduled for revision TKA, simultaneous bilateral knee arthroplasty, unicompartmental knee arthroplasty, or knee arthroplasty 100 for cancer or rheumatic disease. A total of 10 patients were excluded because they met one or more exclusion criteria. A total of 250 met inclusion criteria: 129 declined participation after and 121 provided written informed consent and were included in this study."

Author response: The suggestion is very much appreciated and incorporated into the paper.

Changes to manuscript: Please see lines 102-110 on page 5. “This was a secondary analysis of a prospective longitudinal study patients scheduled for a unilateral TKA for osteoarthritis at a hospital in Savannah, Georgia, USA. All adult patients aged 45-90 years, diagnosed with symptomatic knee osteoarthritis (OA), and scheduled for primary TKA by one of nine participating surgeons were considered for inclusion, regardless of race or gender. Participants were excluded if they were scheduled for revision TKA, simultaneous bilateral knee arthroplasty, unicompartmental knee arthroplasty, or knee arthroplasty for cancer or rheumatic disease. A total of 10 patients were excluded because they met one or more exclusion criteria. A total of 250 met inclusion criteria: 129 declined participation after and 121 provided written informed consent and were included in this study.”

- Apart from baseline characteristics, it is important to add the chief complaints of patients/the reason for TKA. As this alone might alter the expectation, pain and outcomes...

Author response: This is an excellent suggestion though these data were not collected beyond completion of the KOOS scales so we are unable to provide this information. With this said, it seems clear that complaints of pain with activity and compromised function are likely subsumed into the KOOS scale ratings. We have noted this in our limitations section. 

Changes to manuscript: Please see lines 342-346 on page 14. “We did not specifically determine participants’ chief complaints but it is likely that the great majority sought TKA because of knee pain and compromised function, and both are captured by KOOS scale scores [48]. “

- line 108: would omit Virgina Commonwealth University. Replace by "the university Institutional Review Boards".

Author response: There is only one university IRB and we would prefer to keep the name of the institution in case there is any question of who approved the review. 

Changes to manuscript: No changes made. 

- line 118: please if possible cite a source for the "10% to 15% average worsening expected without TKA" as this is a big claim.

Author response: Thank you. We have provided two references to support this estimate. 

Changes to manuscript: Please see lines 130-131 on page 6. “…10% to 15% average worsening expected without TKA, and given the costs and risks of surgery [18,19].”

- line 124-127: excellent explanation

Author response: thank you!

- line 146-147: annual income and educational level are mentioned. Consider explaining why these were collected/if looking at SES, etc.

Author response: They were included to provide a thorough description of sample characteristics. This point has been added to the paper. 

Changes to manuscript: Please see line 157 on page 7. “To thoroughly describe our sample,…”

- line 162: regarding the PASS, it remains unclear to the reader, without going back to the reference, whether the PASS only consists of one question (lines 163-165) or it it's a full questionnaire (like mentioned in line 163). Please clarify.

Author response: We have added a brief statement to clarify that this was a single item.

Changes to manuscript: Please see line 175 on page 8. “The single item PASS question was…”

Comment #4: Data analysis

- line 167: why was the study powered to detect a 50th percentile estimate only. Why not 75th as per the KOOS measure in other publications. Please justify/clarify.

Author response: We wanted to reasonably concise estimate for the average patient. This rationale was added to the paper. 

Changes to manuscript:

- line 169-171: why was the study not powered?

Author response: The study was powered for baseline scores but not for follow-up scores because we were unclear about the rate of follow-up and the baseline scores were considered the most critical for SWE estimation.

Changes to manuscript: Please see lines 179-184 on page 8. “ Our study was originally powered to detect a 50th percentile estimate for the SWE. A sample size of 120 would result in a 95% confidence interval of 0.40 to 0.60 which we believed to be reasonable for estimating the 50th percentile for the SWE for the average patient. The study was not powered to establish precision for the six-month postoperative KOOS measures reported on in the current study because it was the baseline scores that were most critical for SWE estimation.”

- lines 177-182: the correct tests were chosen and the statistical analyses serve the purpose of this s

---

## [Decision Letter · Decision Letter 1]

5 Feb 2024

PONE-D-23-35725R1Six-Month Pain and Function Outcome Expectations Were Established for Total Knee Arthroplasty Using the Smallest Worthwhile EffectPLOS ONE

Dear Dr. Riddle,

Thank you for submitting your manuscript to PLOS ONE. After careful consideration, we feel that it has merit but does not fully meet PLOS ONE’s publication criteria as it currently stands. Therefore, we invite you to submit a revised version of the manuscript that addresses the points raised during the review process.

We look forward to receiving your revised manuscript.

Kind regards,

Hassan Zmerly, MD PhD

Academic Editor

PLOS ONE

Journal Requirements:

Reviewers' comments:

Reviewer's Responses to Questions

**Comments to the Author**

1. If the authors have adequately addressed your comments raised in a previous round of review and you feel that this manuscript is now acceptable for publication, you may indicate that here to bypass the “Comments to the Author” section, enter your conflict of interest statement in the “Confidential to Editor” section, and submit your "Accept" recommendation.

Reviewer #1: All comments have been addressed

Reviewer #2: (No Response)

2. Is the manuscript technically sound, and do the data support the conclusions?

Reviewer #1: Yes

Reviewer #2: Yes

3. Has the statistical analysis been performed appropriately and rigorously? 

Reviewer #1: (No Response)

Reviewer #2: Yes

4. Have the authors made all data underlying the findings in their manuscript fully available?

Reviewer #1: (No Response)

Reviewer #2: Yes

5. Is the manuscript presented in an intelligible fashion and written in standard English?

Reviewer #1: (No Response)

Reviewer #2: Yes

6. Review Comments to the Author

Reviewer #1: (No Response)

Reviewer #2: - Line 157: "to thoroughly describe our sample" is not a good justification. This could be said for any variables people recruit. It would be better to try to justify the importance of income and SES.

- Line 246: Expectations Assessment (Not Discussions)

- Good overall

7. PLOS authors have the option to publish the peer review history of their article (what does this mean?). If published, this will include your full peer review and any attached files.

Reviewer #1: **Yes: **VALENTINA DI GREGORI, MD

Reviewer #2: No

---

## [Author Response · Author response to Decision Letter 1]

5 Feb 2024

Responses to review team

The only comments we found were from reviewer #2. Reviewer #1 appeared to be satisfied with the R1 version and had no comments and noted “All comments have been addressed.”

Reviewer #2 had the following 2 comments:

Reviewer #2: - Line 157: "to thoroughly describe our sample" is not a good justification. This could be said for any variables people recruit. It would be better to try to justify the importance of income and SES.

Author response: We are not clear why R2 is concerned about inclusion of income and educational level in our description of sample characteristics. These socially driven variables provide a richer description beyond simple demographics and psychological characteristics but with this said, these variables were not used in the analyses so we have deleted them from the paper. 

Changes to manuscript: We have deleted the income and educational content from the paper. This section now starts with: “Age in years and biological sex were recorded. Depressive symptoms were…”

- Line 246: Expectations Assessment (Not Discussions)

Change to manuscript: We have changed this heading to “Expectations Assessment”

- Good overall

Thank you.

---

## [Editor Report · Decision Letter 2]

28 Feb 2024

Six-Month Pain and Function Outcome Expectations Were Established for Total Knee Arthroplasty Using the Smallest Worthwhile Effect

PONE-D-23-35725R2

Dear Dr. Riddle,

We’re pleased to inform you that your manuscript has been judged scientifically suitable for publication and will be formally accepted for publication once it meets all outstanding technical requirements.

Kind regards,

Hassan Zmerly, MD PhD

Academic Editor

PLOS ONE